# REG: A Regularization Optimizer for Robust Training Dynamics

## Abstract

Optimizers are crucial for the efficient training of Large Language Models (LLMs). While AdamW is the de facto standard, recent structure-aware optimizers like Muon have emerged, which regularize gradient updates by operating on entire weight matrices. The Muon optimizer balances the gradient updates along all the directions. However, Muon's reliance on the matrix sign function can lead to training instability, exhibits incompatibility when fine-tuning models pre-trained with AdamW. To address these limitations, we propose **REG**, a novel optimizer that replaces Muon's aggressive matrix sign operator with the Row-and-Column-Scaling (RACS) operator. Theoretically grounded in balancing a matrix, the RACS operator regularizes the update steps in a less drastic manner, making it simpler to implement and more compatible with established training dynamics. Through extensive empirical experiments on LLM training, we demonstrate that our REG optimizer not only achieves superior performance and stability over AdamW, but also maintains consistency with the AdamW training paradigm. This consistency is particularly evident during the fine-tuning stage, where REG optimizer avoids the performance degradation observed with Muon.

## 1 Introduction

The rapid advancements of Large Language Models (LLMs) (Achiam et al., 2023; Guo et al., 2025; Team et al., 2023) have made training efficient and effective optimizers a critical area of research. While Adam (Kingma & Ba, 2017) and AdamW (Loshchilov & Hutter, 2019) remain standard, recent empirical studies have revealed a key challenge in large-scale LLM training: the momentum matrices within optimizers often become ill-conditioned. This indicates that a few principal directions dominate the parameter updates, which can hinder convergence and stability. This observation has motivated a new class of optimizers, such as Muon (Jordan et al., 2024) and GaLore (Zhao et al., 2024), that explicitly address the structural properties of parameter matrices.

Among these, the Muon optimizer is notable for its unique approach of treating weights as matrices and applying a matrix sign function to orthogonalize the momentum-averaged gradient. While this method successfully addresses the ill-conditioning problem by reducing the spectral condition number, it introduces significant implementation complexity and has been observed to cause training instability. Specifically, the Muon optimizer's aggressive rescaling of singular values can lead to crashes. Furthermore, its training dynamics are not fully consistent with those of AdamW, which can cause performance degradation when fine-tuning an AdamW-trained model. These drawbacks highlight a need for a more stable and computationally simple alternative that can still regularize the update dynamics.

In this paper, we propose a regularized gradient descent with momentum optimizer, dubbed as **REG**. Our approach is grounded in the observation that ill-conditioned momentum matrices can be improved by an operator that makes their rows and columns more uniform in magnitude. Instead of the computationally complex matrix sign function used in Muon, we propose using a Row-and-Column-Scaling (RACS) operator. The RACS operator, which involves simple diagonal matrix multiplications, is computationally efficient and straightforward to implement. We provide theoretical grounding for this approach by drawing on classic results from numerical analysis, which show that row and column scaling can significantly improve a matrix's conditioning. Critically, we demonstrate that the RACS operator provides a less drastic regularization than the matrix sign

function, resulting in a training process that is more stable and compatible with AdamW-trained models.

Our final proposed algorithm integrates this RACS operator into the standard Gradient Descent with Momentum (GDM) framework, along with two practical enhancements: weight decay to prevent overfitting and an RMS-based rescaling to ensure consistent update magnitudes. The latter is particularly robust for the empirically superior choice of $\ell_2$-norm scaling, where we derive a closed-form solution for the RMS of the normalized matrix.

Our contributions are summarized as follows:

- We propose a novel optimizer, named REG, which regularizes the update steps using the computationally efficient and stable RACS operator.

- We provide a theoretical justification for using the RACS operator by connecting it to classic results on matrix equilibration, and we provide a closed-form expression for the RMS of the update matrix when using $\ell_2$-norm normalization.

- Through a series of empirical experiments, we demonstrate that our optimizer achieves superior performance for LLM training while offering greater stability and consistency with the AdamW training paradigm compared to Muon, particularly during the fine-tuning stage.

## 2 RELATED WORK

**Optimizers**   The development of optimization algorithms for training machine learning models has seen significant progress since the introduction of stochastic gradient descent with momentum (SGDM) by Polyak (1964). Subsequent research has led to a multitude of advanced optimizers, including but not limited to those proposed by Dozat (2016); Duchi et al. (2011); Graves (2013); Loshchilov & Hutter (2019); Kingma & Ba (2017); Martens (2020); Shazeer & Stern (2018); Zeiler (2012). Among these, Adam (Kingma & Ba, 2017) and its variant with decoupled weight decay, AdamW (Loshchilov & Hutter, 2019), have become the de facto standard for training large language models (LLMs). Unlike SGDM, the Adam family of optimizers employs adaptive learning rates for each parameter, allowing the training process to more effectively navigate the objective function landscape, especially in regions with varying curvature. However, the use of first and second momentum terms in Adam introduces a considerable memory overhead.

To address this memory challenge, several works have focused on developing more memory-efficient and accelerated variants of Adam. Adafactor (Shazeer & Stern, 2018) reduces memory usage by omitting the first momentum term and approximating the second momentum term using a factored representation inspired by the divergence method (Lee & Seung, 1999). Similarly, Anil et al. (2019) proposed a memory-efficient variation of Adagrad (Duchi et al., 2011). Another successful strategy involves the use of low-rank approximation. The LoRA method (Hu et al., 2022) facilitates the efficient fine-tuning of LLMs by training low-rank matrices $A$ and $B$ instead of the full weight matrix $W$. Extending this concept to the optimizer itself, GaLore (Zhao et al., 2024) modifies the Adam optimizer by replacing the full gradient with its low-rank approximation, thereby reducing the memory footprint of both the first and second momentum terms. A related approach, Flora (Hao et al., 2024), is based on a similar principle. In a different vein, the Muon optimizer (Jordan et al., 2024) has shown promising results in LLM training (Liu et al., 2025; Team et al., 2025). This approach suggests that an optimizer should balance the update matrix to ensure that all parameters are updated along all directions. While the original Muon optimizer faced challenges with training very large LLMs, its variants, such as those discussed by Team et al. (2025), have achieved success in training models with up to 1 trillion parameters.

**Matrix Balancing**   Matrix balancing is a well-established problem in numerical analysis, traditionally studied for its application in the numerical solution of linear equations (Hildebrand, 1987; Horn & Johnson, 2012). The Row-And-Column-Scaling (RACS) operator, as discussed in works such as (Bauer, 1963; Van der Sluis, 1969), is a widely used method for this purpose. Research by Bauer (1963); Forsythe & Straus (1955); Van der Sluis (1969); Yang et al. (2024) has explored the effects of RACS on the condition numbers and other properties of matrices. The RACS operator

also finds application in solving optimization problems, particularly in balancing matrices within primal-dual formulations, as demonstrated by Ruiz (2001); Pock & Chambolle (2011).

## 3 ALGORITHM

### 3.1 MOTIVATION

We consider the optimization problem of minimizing a differentiable function $f : \mathbb{R}^{m \times n} \to \mathbb{R}$ over a parameter matrix $W \in \mathbb{R}^{m \times n}$. A foundational and widely-adopted method for this class of problems is Gradient Descent with Momentum (GDM) (Polyak, 1964). At each iteration $k$, given the current parameter matrix $W_k$, the momentum matrix $M_k$, a learning rate $\alpha$, and a momentum coefficient $\mu$, the GDM update rules are defined as:

$$
\begin{aligned}
M_{k+1} &= \mu M_k + (1 - \mu) \nabla f(W_k), \\
W_{k+1} &= W_k - \alpha M_{k+1}.
\end{aligned}
\tag{1}
$$

Recent empirical studies on LLMs have revealed that for 2D parameter matrices within Transformer architectures, the corresponding momentum matrix $M$ is frequently observed to be ill-conditioned, exhibiting a large spectral condition number ($\sigma_{\max}/\sigma_{\min}$) (Gupta et al., 2018; Jordan et al., 2024; Zhao et al., 2024). A high spectral condition number indicates that the matrix's energy is concentrated along a few principal directions, which in turn implies that parameter updates are dominated by these directions. This suggests the presence of an intrinsically low-rank structure within the update dynamics.

This observation motivates the introduction of a computationally efficient regularization operator, denoted by $\mathrm{reg}(\cdot)$, applied to the momentum matrix $M_k$ with the objective of improving its conditioning. We thus propose the regularized GDM optimizer, which incorporates this regularization step into the standard GDM framework:

$$
\begin{aligned}
M_{k+1} &= \mu M_k + (1 - \mu) \nabla f(W_k), \\
M_{k+1} &= \mathrm{reg}(M_{k+1}), \\
W_{k+1} &= W_k - \alpha M_{k+1}.
\end{aligned}
\tag{2}
$$

The choice of the regularization operator $\mathrm{reg}(\cdot)$ is critical. For instance, the Muon optimizer employs the matrix sign function, which theoretically reduces the spectral condition number of the resulting matrix to one (Jordan et al., 2024). In this work, we investigate the RACS operator. Let $\mathcal{D}_k$ denote the set of non-singular diagonal $k \times k$ matrices. The RACS operator is defined as $\mathrm{reg}(M; D_1, D_2) = D_1 M D_2$ for specifically chosen matrices $D_1 \in \mathcal{D}_m$ and $D_2 \in \mathcal{D}_n$. The central problem is to determine appropriate diagonal matrices $D_1$ and $D_2$ to improve the matrix's properties, specifically by minimizing a measure of its ill-conditioning.

The problem of optimal diagonal scaling to improve a matrix's conditioning is a classic topic in numerical analysis. For this, we recall the following foundational result on matrix equilibration established by Van der Sluis (1969).

**Theorem 1** *Let $M \in \mathbb{R}^{m \times n}$ be a matrix. In the following, the norm $\| \cdot \|^*$ may be any Hölder norm or the Frobenius norm.*

*(a) If $\kappa(M) := \|M\|_\infty / \|M\|^*$, then $\kappa(DM)$ is minimal if all rows in $DM$ have equal 1-norm.*

*(b) If $\kappa(M) := \|M\|_1 / \|M\|^*$, then $\kappa(MD)$ is minimal if all columns in $MD$ have equal 1-norm.*

*(c) If $M$ is invertible and $\kappa(M) := (\max_{i,j} |M_{ij}|) \|M^{-1}\|^*$, then $\kappa(DM)$ is minimal if all rows in $DM$ have equal $\infty$-norm.*

*(d) If $M$ is invertible and $\kappa(M) := (\max_{i,j} |M_{ij}|) \|M^{-1}\|^*$, then $\kappa(MD)$ is minimal if all columns in $MD$ have equal $\infty$-norm.*

The functions $\kappa(\cdot)$ in Theorem 1, while distinct from the standard spectral condition number ($\sigma_{\max}(M)/\sigma_{\min}(M)$), serve a conceptually analogous purpose: they quantify the "imbalance" of a matrix. This notion is intrinsically linked to modern concepts of a matrix's effective dimensionality, such as the stable rank ($\|M\|_F^2 / \|M\|_2^2$) and effective rank. A matrix with a low stable rank,

for instance, has its energy concentrated in a few dominant singular vectors, a characteristic that often manifests as rows or columns with disproportionately large norms. The process of "equilibration"—scaling rows and columns to have uniform norms—directly counteracts this imbalance. Therefore, minimizing $\kappa(\cdot)$ via equilibration can be interpreted as a computationally tractable proxy for improving the matrix's effective properties, pushing it towards a state where its constituent rows and columns are more uniform in magnitude. This principle provides a robust theoretical foundation for utilizing row or column normalization as a regularization strategy.

Inspired by these findings, we propose a normalization operator, $\mathrm{normal}(\cdot; p)$, which, for a given matrix $M \in \mathbb{R}^{m \times n}$, normalizes either its rows or columns based on their $\ell_p$-norm. The choice of axis is determined by the matrix dimensions to minimize computational overhead:

$$\mathrm{normal}(M; p) = \begin{cases} \mathrm{diag}(\|M_{1,:}\|_p^{-1}, \ldots, \|M_{m,:}\|_p^{-1})M & \text{if } m \leq n, \\ M\mathrm{diag}(\|M_{:,1}\|_p^{-1}, \ldots, \|M_{:,n}\|_p^{-1}) & \text{if } m > n, \end{cases} \tag{3}$$

where $M_{i,:}$ denotes the $i$-th row of $M$ and $M_{:,j}$ denotes the $j$-th column of $M$. This leads to a regularized optimizer, parameterized by the norm order $p$:

$$\begin{aligned} M_{k+1} &= \mu M_k + (1 - \mu)\nabla f(W_k), \\ M_{k+1} &= \mathrm{normal}(M_{k+1}; p), \\ W_{k+1} &= W_k - \alpha M_{k+1}. \end{aligned} \tag{4}$$

The theoretical results on matrix equilibration primarily support the use of $p = 1$ or $p = \infty$. However, our empirical investigations, particularly in the context of training LLMs, indicate that $p = 2$ yields superior performance. This discrepancy highlights a known gap between classical numerical linear algebra theory and the complex dynamics of deep learning optimization. A comprehensive ablation study to determine the optimal order $p$ across diverse tasks is beyond the scope of this work. Instead, we focus our experimental validation on the SFT of LLMs, where the effectiveness of the $p = 2$ case is demonstrated.

## 3.2 PRACTICAL ENHANCEMENTS FOR LARGE-SCALE TRAINING

To enhance the practical applicability and robustness of the proposed optimizer, particularly for large-scale models, we incorporate two established techniques, following the methodology of recent work Liu et al. (2025).

**Weight Decay** The first enhancement is the inclusion of weight decay, a standard regularization technique in deep learning. It is implemented by adding a term proportional to the current weights $W_k$ to the update rule. This penalizes large weight values, which helps to prevent overfitting and improve generalization.

**Consistent Update Magnitude** A second, more critical, modification addresses the need for consistent update magnitudes. The naive regularization in Equation 4 normalizes the rows or columns of the momentum matrix $M_{k+1}$, but does not control its overall scale. The magnitude of the update could therefore vary unpredictably. Following Liu et al. (2025), we rescale the normalized momentum matrix such that its root mean square falls within a predefined target range (e.g., 0.2 to 0.4).

For a generic $p$-norm, the RMS of the normalized matrix $\mathrm{normal}(M; p)$ does not have a simple closed-form expression. However, for the empirically superior case where $p = 2$, a closed-form solution exists.

**Theorem 2** *For any matrix $M \in \mathbb{R}^{m \times n}$, the root mean square of the $\ell_2$-normalized matrix $\tilde{M} := \mathrm{normal}(M; 2)$ is given by $\sqrt{\frac{1}{\max\{m,n\}}}$.*

**Proof 1** *The proof follows directly from the definitions. Without loss of generality, assume $m \leq n$, which implies normalization is performed row-wise. The case $m > n$ is analogous. The squared RMS of $\tilde{M}$ is:*

$$\mathrm{RMS}(\tilde{M})^2 = \frac{1}{mn}\sum_{i=1}^{m}\sum_{j=1}^{n}\tilde{M}_{ij}^2 = \frac{1}{mn}\sum_{i=1}^{m}\sum_{j=1}^{n}\left(\frac{M_{ij}}{\|M_{i,:}\|_2}\right)^2 = \frac{1}{mn}\sum_{i=1}^{m}\frac{\sum_{j=1}^{n}M_{ij}^2}{\|M_{i,:}\|_2^2} = \frac{1}{n}.$$

*Thus,* $\mathrm{RMS}(\tilde{M}) = \sqrt{\frac{1}{n}}$. *Since we assumed* $m \leq n$, *we have* $n = \max\{m, n\}$, *which completes the proof.*

## 3.3 THE FINAL REGULARIZED OPTIMIZER

By integrating the aforementioned components, we arrive at the final version of our REG optimizer. At each iteration $k$, the update rules are defined as follows:

$$
\begin{aligned}
M_{k+1} &= \mu M_k + (1 - \mu)\nabla f(W_k), \\
\tilde{M}_{k+1} &= \mathrm{normal}(M_{k+1}; p), \\
\hat{M}_{k+1} &= \tilde{M}_{k+1} \cdot \frac{\rho_{\mathrm{target}}}{\mathrm{RMS}(\tilde{M}_{k+1})}, \\
W_{k+1} &= W_k - \alpha(\hat{M}_{k+1} + \lambda W_k),
\end{aligned}
\tag{5}
$$

where $\lambda$ is the weight decay coefficient and $\rho_{\mathrm{target}}$ is a hyperparameter representing the target RMS of the update matrix. For the specific case of $p = 2$, the denominator $\mathrm{RMS}(\tilde{M}_{k+1})$ can be replaced by its deterministic value from Theorem 2.

## 4 THEORETICAL CONVERGENCE ANALYSIS

In this section, we present a theoretical convergence analysis of our regularized optimizer. A direct convergence proof for the full algorithm, as defined in equation 5, is highly non-trivial and remains an open problem in numerical optimization. Consequently, our analysis is restricted to a simplified variant of the regularized optimizer, denoted as the naive version and given by equation 4. While a comprehensive theoretical guarantee for the full algorithm is beyond the scope of this work, the analysis presented here provides foundational insights into the convergence behavior of our method.

The analysis in this section focuses on the case where the momentum parameter is set to zero, i.e., $\mu = 0$. The convergence analysis for the case where $\mu \neq 0$ is significantly more complex, as the normalization operator $\mathrm{normal}$ breaks the linearity between the gradient term $\nabla f(W)$ and the momentum term $M$. This nonlinearity prevents the direct application of classical momentum analysis techniques. Nevertheless, the convergence proof for the $\mu = 0$ case establishes a crucial theoretical foundation for our method.

**Theorem 3** *Assume that* $f : \mathbb{R}^{m \times n} \to \mathbb{R}$ *is continuously differentiable and its gradient* $\nabla f$ *is* $L$-*Lipschitz. Consider the iteration equation 4 with* $\mu = 0$. *For a sufficiently small learning rate* $\alpha$ *that depends only on the dimensions* $m$ *and* $n$, *the sequence of gradients converges to zero in Frobenius norm:*

$$
\lim_{k \to \infty} \|\nabla f(W_k)\|_F = 0.
$$

The following theorem demonstrates that the proposed algorithm converges to a stationary point, albeit within a neighborhood. The size of this neighborhood is explicitly determined by the step size $\alpha$ and the momentum parameter $\mu$. Specifically, a smaller step size $\alpha$ and a momentum parameter $\mu$ closer to zero can lead to a tighter bound on the limit inferior of the sum of row norms of the gradient.

**Theorem 4 (Convergence of Row-Normalized Gradient Descent with Momentum)** *Let the following assumptions hold:*

**Assumption 1 (L-smoothness)** *The objective function* $f : \mathbb{R}^{m \times n} \to \mathbb{R}$ *is* $L$-*smooth with respect to the Frobenius norm, i.e., for any* $X, Y \in \mathbb{R}^{m \times n}$, *we have*

$$
\|\nabla f(X) - \nabla f(Y)\|_F \leq L \|X - Y\|_F
$$

**Assumption 2 (Bounded Below)** *The function* $f$ *is bounded below by a scalar* $f^* > -\infty$.

*Consider the algorithm with $p = 2, m \le n$, defined for $k \ge 0$ as:*

$$M'_{k+1} = \mu M_k + (1 - \mu)\nabla f(W_k),$$

$$M_{k+1} = \text{normal}(M'_{k+1}; 2) = \text{diag}(\left\| (M'_{k+1})_{1,:} \right\|_2^{-1}, \ldots, \left\| (M'_{k+1})_{m,:} \right\|_2^{-1}) M'_{k+1},$$

$$W_{k+1} = W_k - \alpha M_{k+1},$$

*where $0 < \mu < 1$, $\alpha > 0$, and we initialize $M_0 = \mathbf{0}$. We assume $(M'_{k+1})_{i,:} \ne \mathbf{0}$ for all $i, k$, ensuring the normalization is well-defined. Let $g_k = \sum_{i=1}^{m} \left\| (\nabla f(W_k))_{i,:} \right\|_2$.*

*Then, the limit inferior of the sum of row norms of the gradient is bounded as:*

$$\liminf_{k \to \infty} g_k \le \frac{L\alpha m}{2} + \frac{2\mu m}{1 - \mu}$$

While $g_k$ is not the standard Frobenius norm of the gradient matrix, it serves as a useful proxy. The Frobenius norm of the gradient, $\|\nabla f(W_k)\|_F$, can be bounded by $g_k$ given that the $L_2$ norm of each row of the normalized matrix $M_{k+1}$ is unity. However, we do not have a direct theoretical guarantee on the convergence of $\|\nabla f(W_k)\|_F$ to zero in the presence of momentum ($\mu \ne 0$), hence the result is expressed in terms of an upper bound on the gradient's limit inferior. A full theoretical proof for this general case is beyond the scope of this paper, and we leave a more rigorous analysis to future work. The provided theorem offers a foundational understanding of the algorithm's behavior, showing that it does not diverge and approaches a region of low gradient.

## 5 EXPERIMENTS

### 5.1 SFT WITH FULL-PARAMETERS

**Math Word Problems**  In this part, we conduct an experimental evaluation of mathematical reasoning capabilities. Specifically, we fine-tune the Qwen2.5-Math-1.5B model (Yang et al., 2024) using a 20K data subset sampled from the NuminaMath-CoT dataset (LI et al., 2024). We compare our proposed REG optimizer with three established baselines: the standard AdamW (Loshchilov & Hutter, 2019), Muon (Jordan et al., 2024), and NGD (Newtonian Gradient Descent). The performance of the fine-tuned models is evaluated on several downstream mathematical reasoning tasks, including GSM8K (Cobbe et al., 2021), MATH500 (Lightman et al., 2023), and AIME24.

The AIME24 dataset, consisting of only 30 problems, is noted to yield unstable results with high variance; therefore, readers should focus on the more robust and statistically significant results from the GSM8K and MATH500 benchmarks. All experiments were conducted using the EvalScope framework (Team, 2024). The test accuracies for models fine-tuned with different optimizers are summarized in Table 1.

| Optimizers | GSM8K(%) | MATH500(%) | AIME24(%) | Average(%) |
|---|---|---|---|---|
| Qwen2.5-Math-1.5B | 28.1 | 24.4 | **13.3** | 21.9 |
| Muon | 49.1 | 58.6 | 10.0 | 39.2 |
| NGD | 70.1 | 60.2 | 6.7 | 45.6 |
| AdamW | **77.8** | 61.8 | 10.0 | 49.8 |
| **REG(ours)** | 76.5 | **64.8** | 10.0 | **50.4** |

Table 1: Test Accuracies of fine-tuned models on downstream mathematical reasoning tasks. REG optimizer achieves competitive and, in some cases, superior performance compared to AdamW and other baselines.

As shown in Table 1, our proposed REG optimizer achieves highly competitive performance, surpassing AdamW in terms of average accuracy. Specifically, REG optimizer obtains a remarkable 64.8% accuracy on MATH500, outperforming all other optimizers. On the GSM8K benchmark, REG optimizer's performance (76.5%) is on par with the leading AdamW (77.8%), with a marginal difference that is not statistically significant.

In contrast, the Muon optimizer exhibits a significant performance degradation on the GSM8K task, achieving only 49.1% accuracy, which is considerably lower than both AdamW and our method. Our method's ability to maintain high performance across multiple tasks demonstrates its robustness and effectiveness for mathematical fine-tuning.

**Mathematical Optimization Modeling Problem**   We conduct an empirical experiment on mathematical optimization modeling problems (Ramamonjison et al., 2023; Huang et al., 2024; 2025). This task aims to generate solvable mathematical models from a natural language description of an optimization problem, and then use a solver to find the optimal solution. We use Qwen3-4B-Instruct-2507 (Team, 2025) as the base model and fine-tune it using a mixed training dataset. The model's performance is evaluated on several standard benchmarks: MAMO (Huang et al., 2024), NL4OPT (Ramamonjison et al., 2023), IndustryOR-fixed (Xiao et al., 2025), and OptMATH-Bench (Lu et al., 2025). The training results comparing different optimizers are presented in Table 2.

| | MAMO | | NL4OPT | IndustryOR | OptMATH-Bench | Average |
|---|---|---|---|---|---|---|
| | EasyLP | ComplexLP | | | | |
| Qwen3-4B | 74.08 | 22.28 | 82.04 | 27.00 | 9.84 | 43.05 |
| AdamW | 83.59 | 33.18 | 86.12 | 36.00 | 4.15 | 48.61 |
| Muon | 84.66 | 33.65 | 83.26 | 37.00 | 5.18 | 48.75 |
| **REG(ours)** | **84.66** | **35.55** | **87.35** | **37.00** | **11.40** | **51.19** |

Table 2: Performance comparison of different optimizers on mathematical optimization modeling tasks. "MAMO" is grouped with two sub-columns: EasyLP and ComplexLP. Other datasets occupy single columns with vertically merged headers.

As shown in Table 2, fine-tuning with any optimizer significantly improves the model's performance across all datasets compared to the original Qwen3-4B-Instruct-2507 model. Our REG optimizer consistently achieves the highest accuracy on most benchmarks. Specifically, it outperforms all other optimizers on NL4OPT, MAMO-ComplexLP, and OptMATH-Bench, while matching the best performance on MAMO-EasyLP and IndustryOR. The most notable improvement is observed on the challenging OptMATH-Bench dataset, where our method achieves 11.40% accuracy, more than doubling the performance of AdamW and Muon. The average accuracy across all five datasets further highlights the superiority of our approach, with REG optimizer scoring 51.19%, compared to 48.61% for AdamW and 48.75% for Muon. This demonstrates the effectiveness of REG optimizer in enhancing the model's ability to handle complex and diverse mathematical optimization problems.

## 5.2 APPLICATION TO IMAGE CLASSIFICATION

Having validated the REG optimizer on natural language processing benchmarks, we now extend our evaluation to the domain of computer vision. To this end, we assess its performance on the CIFAR-100 image classification task (Krizhevsky et al., 2009), which contains 60,000 color images across 100 classes. Specifically, we train ResNet-18 and ResNet-50 models (He et al., 2016) from scratch using the REG optimizer for parameter updates. We use SGD, NGD, and Adam as baseline optimizers. We select Adam over AdamW as it is often the preferred choice for computer vision tasks. The results of this experiment are summarized in Table 3, and the training curves for loss and accuracy are presented in Figure 1.

| Models | SGD | NGD | Adam | **REG(ours)** |
|---|---|---|---|---|
| ResNet-18 (%) | 41.37 | 22.43 | 58.65 | **59.03** |
| ResNet-50 (%) | 33.04 | 13.01 | **59.62** | 59.14 |

Table 3: Test accuracies on the CIFAR-100 dataset for various optimizers and models.

As shown in Table 3, the REG optimizer achieves superior performances among all the optimizers.This indicates that while Adam remains a very strong baseline, the adaptive rescaling mechanism of our method is highly effective compared to non-adaptive or poorly-formulated second-order approaches like NGD.

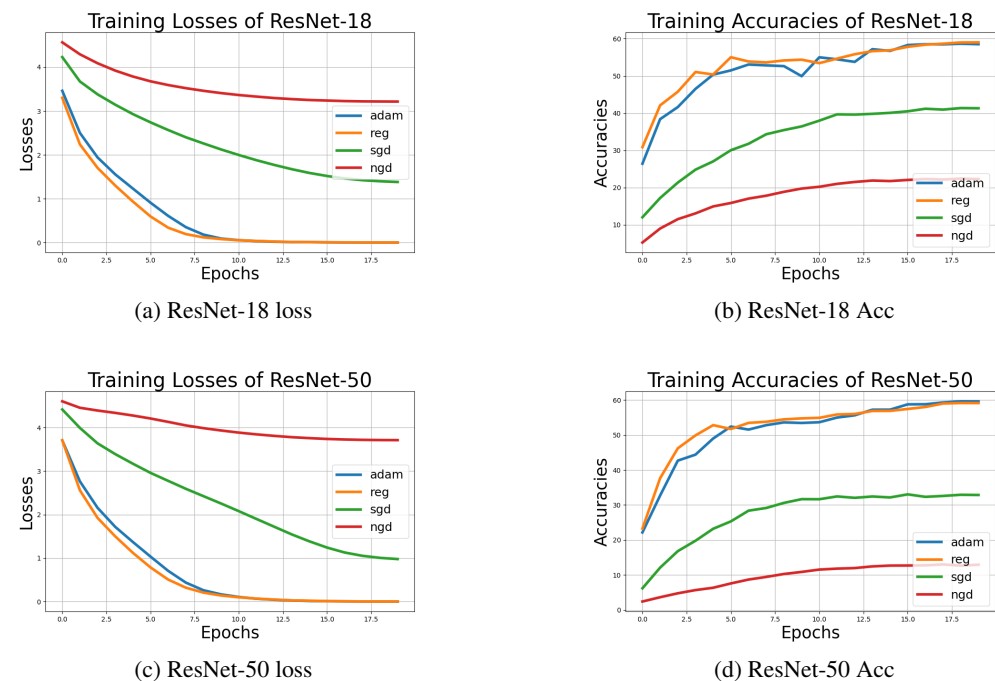

Figure 1: Training loss and accuracy curves on the CIFAR-100 image classification task.

A more detailed analysis of the training curves in Figure 1 reveals a key advantage of our optimizer. The REG optimizer demonstrates the fastest convergence in terms of both loss reduction and accuracy improvement. Notably, its training loss curve descends more rapidly and its training accuracy curve ascends more steeply than all other optimizers. This suggests that the REG optimizer is particularly efficient at quickly locating a good, albeit potentially sub-optimal, parameter configuration early in the training process. This rapid convergence is a desirable property for large-scale training where computational resources are a primary concern, as it allows for the possibility of achieving a reasonable performance with fewer training iterations.

## 5.3 Ablation Studies

**On the Necessity of a Hybrid AdamW Approach.** In this section, we investigate the necessity and efficacy of a hybrid optimization strategy that integrates AdamW updates for specific parameter groups within the **REG** optimization framework. This study is motivated by established practices in similar optimization algorithms, such as Muon, where it is a recommended practice to train embedding layers using AdamW while applying the core optimization algorithm to all other parameters. We hypothesize that this hybrid approach is crucial for mitigating potential numerical instabilities that may arise from applying matrix-based update mechanisms, such as the Newton-Schulz iteration, to the unique structural properties of embedding matrices, which are typically large and sparse.

To empirically validate this hypothesis, we conducted an ablation study comparing two distinct optimizer configurations. The first is the pure **REG** optimizer, which applies its update rule uniformly across all model parameters. The second, designated as **REG-with-AdamW**, is a hybrid variant that employs the AdamW update rule exclusively for the Large Language Model's (LLM) embedding layers, while retaining the **REG** update for all other parameters. A comparative analysis of their performance on a suite of downstream tasks is presented in Table 4, providing a direct assessment of the practical benefits of the proposed hybrid approach.

The results presented in Table 4 demonstrate that the hybrid **REG-with-AdamW** optimizer consistently outperforms the pure **REG** optimizer across a majority of the evaluated tasks, and yields a superior average performance. This empirical evidence supports our hypothesis regarding the necessity of a hybrid strategy and confirms the practical advantage of applying AdamW updates

| Datasets | REG | REG-With-AdamW |
|---|---|---|
| GSM8K (%) | **77.8** | 76.5 |
| MATH500 (%) | 62.4 | **64.8** |
| AIME24 (%) | 3.3 | **10.0** |
| NL4OPT(%) | **88.6** | 87.4 |
| MAMO-EasyLP(%) | **85.6** | 84.7 |
| MAMO-ComplexLP(%) | 28.4 | **35.6** |
| IndustryOR(%) | 26.0 | **37.0** |
| OptMATH-Bench(%) | 7.2 | **11.4** |
| Average (%) | 47.4 | **50.9** |

Table 4: Performance comparison of **REG** and **REG-with-AdamW** on various tasks. The table presents the fine-tuned model's accuracy on a range of datasets.

specifically to the embedding layers while using the **REG** optimizer for the remaining parameters. Consequently, we recommend the adoption of the **REG-with-AdamW** configuration for optimal performance.

**Hyperparameter Selection for Order** $p$. We investigated the selection of the hyperparameter $p$ for Large Language Model (LLM) training. While theoretical guarantees for balancing update matrices exist for $p = 1$ and $p = +\infty$, these values pose a practical challenge: the Root Mean Square (RMS) norm of the update matrix lacks a closed-form solution. This absence complicates the necessary rescaling of updates, thus impacting computational efficiency. Conversely, for $p = 2$, the RMS norm has a closed-form solution, which significantly improves computational efficiency, despite the absence of theoretical guarantees. The practical performance of different hyperparameter choices for $p$ is detailed in Table 5.

| | GSM8K (%) | MATH500 (%) | AIME24 (%) | Average (%) |
|---|---|---|---|---|
| $p = 1$ | 75.8 | 63.6 | 10.0 | 49.8 |
| $p = 2$ | **76.5** | **64.8** | **10.0** | **50.4** |
| $p = +\infty$ | 75.6 | 63.2 | 3.3 | 47.4 |

Table 5: Performance comparison of different hyperparameter $p$ on mathematical tasks. The table shows the fine-tuned model's accuracy on various datasets.

Given the limited sample size of the AIME24 dataset (only 30 problems), we suggest its results be considered with caution. Based on the more extensive experiments conducted on GSM8K and MATH500, we observed that while $p = 2$ lacks a theoretical guarantee, it achieves superior performance compared to the other hyperparameters. This finding highlights an interesting gap between theoretical predictions and empirical reality. Consequently, we recommend using $p = 2$ for training, as it offers a compelling combination of computational efficiency and empirical effectiveness.

## 6 CONCLUSION

In this paper, we introduced a novel optimizer, **REG**, designed to enhance the training of LMs through the use of the RACS operator. We provided a comprehensive analysis of its theoretical underpinnings and validated its effectiveness through extensive empirical experiments. The results consistently demonstrated that the REG optimizer achieves superior performance across diverse tasks, both language and vision tasks. Furthermore, our findings suggest that REG is more aligned with the performance characteristics of AdamW than Muon, indicating its significant potential for future SFT applications.

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

# A  ADDITIONAL PROOFS

**Proof 2 (Proof of Theorem 3)** *Without loss of generality, we assume $m \leq n$. The proof for the case $m \geq n$ is analogous, as the Frobenius inner product is symmetric with respect to transposition, i.e., $\langle A, B \rangle_F = \langle A^\top, B^\top \rangle_F$ for any matrices $A, B \in \mathbb{R}^{m \times n}$.*

**Step 1. Preliminaries and Key Inequality**. *From the $L$-Lipschitz continuity of the gradient $\nabla f$, we have the following property, often known as the Descent Lemma:*

$$f(Y) \leq f(X) + \langle \nabla f(X), Y - X \rangle_F + \frac{L}{2} \|Y - X\|_F^2$$

*where $\langle A, B \rangle_F = \mathrm{Tr}(A^T B)$ is the Frobenius inner product.*

*Let $G_k := \nabla f(W_k)$ and $\tilde{G}_k := \mathrm{normal}(G_k; p)$. By substituting the update rule $W_{k+1} = W_k - \alpha \tilde{G}_k$ into this inequality with $X = W_k$ and $Y = W_{k+1}$, and recalling that $G_k = \nabla f(W_k)$, we obtain:*

$$f(W_{k+1}) \leq f(W_k) + \langle G_k, W_{k+1} - W_k \rangle_F + \frac{L}{2} \|W_{k+1} - W_k\|_F^2$$

$$= f(W_k) - \alpha \langle G_k, \tilde{G}_k \rangle_F + \frac{L\alpha^2}{2} \|\tilde{G}_k\|_F^2 \tag{6}$$

*This inequality is central to our proof. The goal is to show that as long as $G_k \neq 0$, the sum of the last two terms on the right-hand side is negative, guaranteeing that $f(W_k)$ is a strictly decreasing sequence.*

**Step 2. Analysis of the Update Direction** $\tilde{G}_k$. *By definition, for $m \leq n$, the $i$-th row of $\tilde{G}_k$, denoted $(\tilde{G}_k)_i$, is:*

$$(\tilde{G}_k)_i = \frac{(G_k)_i}{\|(G_k)_i\|_p}$$

*(We adopt the convention that if $(G_k)_i = 0$, then $(\tilde{G}_k)_i = 0$).*

*Let's analyze the inner product term $\langle G_k, \tilde{G}_k \rangle_F$ and the norm term $\|\tilde{G}_k\|_F^2$:*

$$\langle G_k, \tilde{G}_k \rangle_F = \sum_{i=1}^m \langle (G_k)_i, (\tilde{G}_k)_i \rangle = \sum_{i=1}^m \frac{\langle (G_k)_i, (G_k)_i \rangle}{\|(G_k)_i\|_p} = \sum_{i=1}^m \frac{\|(G_k)_i\|_2^2}{\|(G_k)_i\|_p}$$

$$\|\tilde{G}_k\|_F^2 = \sum_{i=1}^m \|(\tilde{G}_k)_i\|_2^2 = \sum_{i=1}^m \frac{\|(G_k)_i\|_2^2}{\|(G_k)_i\|_p^2}$$

*Provided $G_k \neq 0$, we have $\langle G_k, \tilde{G}_k \rangle_F > 0$, which confirms that $-\tilde{G}_k$ is a descent direction.*

**Step 3. Bounding the Key Ratio**. *To ensure convergence, the step size $\alpha$ must be chosen carefully. From Eq. equation 6, the decrease in $f$ depends on the relationship between $\langle G_k, \tilde{G}_k \rangle_F$ and $\|\tilde{G}_k\|_F^2$. Let us define their ratio as $\gamma_k$:*

$$\gamma_k = \frac{\|\tilde{G}_k\|_F^2}{\langle G_k, \tilde{G}_k \rangle_F} = \frac{\sum_{i=1}^m \|(G_k)_i\|_2^2 / \|(G_k)_i\|_p^2}{\sum_{i=1}^m \|(G_k)_i\|_2^2 / \|(G_k)_i\|_p}$$

*Let $v_i = (G_k)_i$, $w_i = \|v_i\|_2^2 / \|v_i\|_p > 0$ (for $v_i \neq 0$), and $z_i = \|v_i\|_2 / \|v_i\|_p$. Then $\gamma_k$ can be written as a weighted average of the $z_i$:*

$$\gamma_k = \frac{\sum_{i=1}^m w_i z_i}{\sum_{i=1}^m w_i}$$

*Therefore, the value of $\gamma_k$ must lie between the minimum and maximum values of $z_i$:*

$$\min_{i:(G_k)_i \neq 0} \left( \frac{\|(G_k)_i\|_2}{\|(G_k)_i\|_p} \right) \leq \gamma_k \leq \max_{i:(G_k)_i \neq 0} \left( \frac{\|(G_k)_i\|_2}{\|(G_k)_i\|_p} \right)$$

*In the finite-dimensional vector space $\mathbb{R}^n$, all norms are equivalent. This means there exist positive constants $\delta$ and $\Gamma$, depending only on the dimension $n$ and the choice of norm $p$, such that for any non-zero vector $v \in \mathbb{R}^n$:*

$$0 < \delta \leq \frac{\|v\|_2}{\|v\|_p} \leq \Gamma$$

*Consequently, the ratio $\gamma_k$ is uniformly bounded:*

$$0 < \delta \le \gamma_k \le \Gamma$$

**Step 4. Ensuring Sufficient Decrease in Function Value**. *Rearranging the inequality for $f(W_{k+1})$ from Eq. equation 6:*

$$f(W_{k+1}) \le f(W_k) - \alpha\langle G_k, \tilde{G}_k\rangle_F \left(1 - \frac{L\alpha}{2}\frac{\|\tilde{G}_k\|_F^2}{\langle G_k, \tilde{G}_k\rangle_F}\right) \tag{7}$$

$$= f(W_k) - \alpha\langle G_k, \tilde{G}_k\rangle_F(1 - \frac{L\alpha\gamma_k}{2}) \tag{8}$$

*To guarantee a strict decrease in the sequence $f(W_k)$, we need the term in the parenthesis to be positive. We select a fixed step size $\alpha$ that satisfies this condition for all possible values of $\gamma_k$. Since $\gamma_k \le \Gamma^2$, we must choose $\alpha$ such that:*

$$1 - \frac{L\alpha\Gamma^2}{2} > 0 \implies \alpha < \frac{2}{L\Gamma^2}$$

*Let's choose a step size $\alpha$ such that $0 < \alpha < \frac{2}{L\Gamma^2}$. Let $c = 1 - \frac{L\alpha\Gamma^2}{2}$, which is a positive constant. We then have:*

$$f(W_{k+1}) \le f(W_k) - \alpha(1 - \frac{L\alpha\gamma_k}{2})\langle G_k, \tilde{G}_k\rangle_F \le f(W_k) - c\alpha\langle G_k, \tilde{G}_k\rangle_F$$

*This gives us:*

$$f(W_k) - f(W_{k+1}) \ge c\alpha\langle G_k, \tilde{G}_k\rangle_F$$

**Step 5. Proving the Gradient Norm Converges to Zero**. *Summing the above inequality from $k = 0$ to $N - 1$:*

$$\sum_{k=0}^{N-1}(f(W_k) - f(W_{k+1})) \ge c\alpha\sum_{k=0}^{N-1}\langle G_k, \tilde{G}_k\rangle_F$$

*The left-hand side is a telescoping sum:*

$$f(W_0) - f(W_N) \ge c\alpha\sum_{k=0}^{N-1}\langle G_k, \tilde{G}_k\rangle_F$$

*Since $f$ is bounded below by $f_{inf}$, we have $f(W_N) \ge f_{inf}$. Therefore:*

$$f(W_0) - f_{inf} \ge c\alpha\sum_{k=0}^{N-1}\langle G_k, \tilde{G}_k\rangle_F$$

*As $N \to \infty$, the left-hand side is a finite constant. The right-hand side is the partial sum of a series with non-negative terms. This implies that the series must converge:*

$$\sum_{k=0}^{\infty}\langle G_k, \tilde{G}_k\rangle_F < \infty$$

*A necessary condition for a series to converge is that its general term must approach zero. Thus:*

$$\lim_{k\to\infty}\langle G_k, \tilde{G}_k\rangle_F = \lim_{k\to\infty}\sum_{i=1}^{m}\frac{\|(G_k)_i\|_2^2}{\|(G_k)_i\|_p} = 0$$

*Now, we use norm equivalence again. There exists a constant $\Gamma_p$ such that $\|v\|_p \le \Gamma_p\|v\|_2$.*

$$\sum_{i=1}^{m}\frac{\|(G_k)_i\|_2^2}{\|(G_k)_i\|_p} \ge \sum_{i=1}^{m}\frac{\|(G_k)_i\|_2^2}{\Gamma_p\|(G_k)_i\|_2} = \frac{1}{\Gamma_p}\sum_{i=1}^{m}\|(G_k)_i\|_2$$

*Since $\langle G_k, \tilde{G}_k\rangle_F \to 0$ and each term in the sum is non-negative, we must have:*

$$\lim_{k\to\infty}\sum_{i=1}^{m}\|(G_k)_i\|_2 = 0$$

*This further implies that for each $i = 1, \ldots, m$, $\lim_{k \to \infty} \|(G_k)_i\|_2 = 0$. Consequently, the squared Frobenius norm of the entire gradient matrix also tends to zero:*

$$\lim_{k \to \infty} \|G_k\|_F^2 = \lim_{k \to \infty} \sum_{i=1}^m \|(G_k)_i\|_2^2 = 0$$

*which means $\lim_{k \to \infty} \|\nabla f(W_k)\|_F = 0$.*

*We have shown that under the given assumptions, the norm of the gradient, $\|\nabla f(W_k)\|_F$, converges to 0. By definition of a stationary point, this means that if the sequence $\{W_k\}$ converges to a point $W^*$, then $W^*$ must be a stationary point of $f$ (i.e., $\nabla f(W^*) = 0$).*

The proof of Theorem 4 is tricky. We firstly introduce the auxiliary Lyapunov function. By carefully defining the Lyapunov function, we can prove the final result.

**Proof 3 (Proof of Theorem 4)** *We define a Lyapunov function $L_k$ for $k \geq 0$:*

$$L_k = f(W_k) + c\|M_k\|_F^2$$

*where $c > 0$ is a constant to be determined later. By the definition of the* normal *operator, for any $k \geq 1$, the rows of $M_k$ are unit vectors in the $\ell_2$-norm. Thus, $\|(M_k)_{i,:}\|_2^2 = 1$ for all $i = 1, \ldots, m$. This implies that for $k \geq 1$, $\|M_k\|_F^2 = \sum_{i=1}^m \|(M_k)_{i,:}\|_2^2 = m$. We initialize with $M_0 = \mathbf{0}$, so $\|M_0\|_F^2 = 0$.*

*Let's analyze the one-step change in the Lyapunov function for $k \geq 0$.*

$$L_{k+1} - L_k = (f(W_{k+1}) - f(W_k)) + c(\|M_{k+1}\|_F^2 - \|M_k\|_F^2)$$

*From the $L$-smoothness assumption, we have the descent lemma:*

$$f(W_{k+1}) \leq f(W_k) + \langle \nabla f(W_k), W_{k+1} - W_k \rangle + \frac{L}{2}\|W_{k+1} - W_k\|_F^2$$

$$= f(W_k) - \alpha \langle \nabla f(W_k), M_{k+1} \rangle + \frac{L}{2}\|-\alpha M_{k+1}\|_F^2$$

$$= f(W_k) - \alpha \langle \nabla f(W_k), M_{k+1} \rangle + \frac{L\alpha^2}{2}\|M_{k+1}\|_F^2$$

*Since $\|M_{k+1}\|_F^2 = m$ for $k \geq 0$, this simplifies to:*

$$f(W_{k+1}) - f(W_k) \leq -\alpha \langle \nabla f(W_k), M_{k+1} \rangle + \frac{L\alpha^2 m}{2}$$

*Substituting this into the Lyapunov difference equation yields:*

$$L_{k+1} - L_k \leq -\alpha \langle \nabla f(W_k), M_{k+1} \rangle + \frac{L\alpha^2 m}{2} + c(\|M_{k+1}\|_F^2 - \|M_k\|_F^2) \qquad (9)$$

*The key is to relate the inner product term to quantities we can control. From the algorithm's definition, we have $(1 - \mu)\nabla f(W_k) = M'_{k+1} - \mu M_k$. Therefore,*

$$\langle \nabla f(W_k), M_{k+1} \rangle = \frac{1}{1-\mu}\langle M'_{k+1} - \mu M_k, M_{k+1} \rangle$$

$$= \frac{1}{1-\mu}\left(\langle M'_{k+1}, M_{k+1} \rangle - \mu \langle M_k, M_{k+1} \rangle\right)$$

*By the definition of the normalization,*

$$\langle M'_{k+1}, M_{k+1} \rangle = \sum_{i=1}^m \langle (M'_{k+1})_{i,:}, \frac{(M'_{k+1})_{i,:}}{\|(M'_{k+1})_{i,:}\|_2} \rangle = \sum_{i=1}^m \|(M'_{k+1})_{i,:}\|_2 .$$

*Let's denote this term by $h_k$. For the second inner product, we use Young's inequality ($2\langle a, b \rangle \leq \|a\|_F^2 + \|b\|_F^2$):*

$$-\mu \langle M_k, M_{k+1} \rangle \geq -\frac{\mu}{2}(\|M_k\|_F^2 + \|M_{k+1}\|_F^2)$$

Combining these results, we get a lower bound for the inner product:

$$\langle \nabla f(W_k), M_{k+1} \rangle \geq \frac{1}{1-\mu} \left( h_k - \frac{\mu}{2} (\|M_k\|_F^2 + \|M_{k+1}\|_F^2) \right)$$

Now, substitute this back into (9):

$$L_{k+1} - L_k \leq -\frac{\alpha}{1-\mu} \left( h_k - \frac{\mu}{2} (\|M_k\|_F^2 + \|M_{k+1}\|_F^2) \right) + \frac{L\alpha^2 m}{2} + c(\|M_{k+1}\|_F^2 - \|M_k\|_F^2)$$

Rearranging the terms based on $\|M_k\|_F^2$ and $\|M_{k+1}\|_F^2$:

$$L_{k+1} - L_k \leq -\frac{\alpha h_k}{1-\mu} + \left( \frac{\alpha\mu}{2(1-\mu)} - c \right) \|M_k\|_F^2 + \left( \frac{\alpha\mu}{2(1-\mu)} + c \right) \|M_{k+1}\|_F^2 + \frac{L\alpha^2 m}{2}$$

To simplify the expression, we choose the constant $c$ to eliminate the $\|M_k\|_F^2$ term:

$$c = \frac{\alpha\mu}{2(1-\mu)}$$

Since $0 < \mu < 1$ and $\alpha > 0$, we have $c > 0$. With this choice of $c$, the inequality becomes:

$$L_{k+1} - L_k \leq -\frac{\alpha h_k}{1-\mu} + \left( \frac{\alpha\mu}{2(1-\mu)} + \frac{\alpha\mu}{2(1-\mu)} \right) \|M_{k+1}\|_F^2 + \frac{L\alpha^2 m}{2}$$

$$= -\frac{\alpha h_k}{1-\mu} + \frac{\alpha\mu}{1-\mu} \|M_{k+1}\|_F^2 + \frac{L\alpha^2 m}{2}$$

Since $\|M_{k+1}\|_F^2 = m$, we have:

$$L_{k+1} - L_k \leq -\frac{\alpha h_k}{1-\mu} + \frac{\alpha\mu m}{1-\mu} + \frac{L\alpha^2 m}{2} \tag{10}$$

Next, we relate $h_k = \sum_{i=1}^m \left\| (M'_{k+1})_{i,:} \right\|_2$ to the gradient norm sum $g_k = \sum_{i=1}^m \left\| (\nabla f(W_k))_{i,:} \right\|_2$. Using the reverse triangle inequality on the sum of row norms:

$$h_k = \sum_{i=1}^m \left\| (\mu M_k + (1-\mu)\nabla f(W_k))_{i,:} \right\|_2 \geq \sum_{i=1}^m \left( \left\| ((1-\mu)\nabla f(W_k))_{i,:} \right\|_2 - \left\| (\mu M_k)_{i,:} \right\|_2 \right)$$

$$= (1-\mu) \sum_{i=1}^m \left\| (\nabla f(W_k))_{i,:} \right\|_2 - \mu \sum_{i=1}^m \left\| (M_k)_{i,:} \right\|_2$$

$$= (1-\mu)g_k - \mu \sum_{i=1}^m 1 = (1-\mu)g_k - \mu m$$

Note that for $k = 0$, $M_0 = \mathbf{0}$, so $h_0 \geq (1-\mu)g_0$. The inequality $h_k \geq (1-\mu)g_k - \mu m$ holds for all $k \geq 0$. Substituting this lower bound for $h_k$ into (10):

$$L_{k+1} - L_k \leq -\frac{\alpha}{1-\mu}((1-\mu)g_k - \mu m) + \frac{\alpha\mu m}{1-\mu} + \frac{L\alpha^2 m}{2}$$

$$= -\alpha g_k + \frac{\alpha\mu m}{1-\mu} + \frac{\alpha\mu m}{1-\mu} + \frac{L\alpha^2 m}{2}$$

$$= -\alpha g_k + \frac{2\alpha\mu m}{1-\mu} + \frac{L\alpha^2 m}{2}$$

Let $C = \frac{2\alpha\mu m}{1-\mu} + \frac{L\alpha^2 m}{2}$. We have the key recursive inequality:

$$\alpha g_k \leq L_k - L_{k+1} + C$$

Summing from $k = 0$ to $N - 1$:

$$\alpha \sum_{k=0}^{N-1} g_k \leq \sum_{k=0}^{N-1} (L_k - L_{k+1}) + \sum_{k=0}^{N-1} C = (L_0 - L_N) + NC$$

*Let's analyze the boundary terms. $L_0 = f(W_0) + c\|M_0\|_F^2 = f(W_0)$. The Lyapunov function is bounded below because $L_N = f(W_N) + c\|M_N\|_F^2 \geq f^* + c \cdot 0 = f^*$ (since $\|M_N\|_F^2 \geq 0$). Thus, $L_0 - L_N \leq f(W_0) - f^*$.*

$$\alpha \sum_{k=0}^{N-1} g_k \leq f(W_0) - f^* + NC$$

*Dividing by $N\alpha$:*

$$\frac{1}{N} \sum_{k=0}^{N-1} g_k \leq \frac{f(W_0) - f^*}{N\alpha} + \frac{C}{\alpha}$$

*Since the minimum is less than or equal to the average, we have:*

$$\min_{0 \leq k < N} g_k \leq \frac{f(W_0) - f^*}{N\alpha} + \frac{1}{\alpha}\left(\frac{2\alpha\mu m}{1-\mu} + \frac{L\alpha^2 m}{2}\right)$$

$$\min_{0 \leq k < N} g_k \leq \frac{f(W_0) - f^*}{N\alpha} + \frac{2\mu m}{1-\mu} + \frac{L\alpha m}{2}$$

*Taking the limit as $N \to \infty$, the first term vanishes:*

$$\liminf_{k \to \infty} g_k \leq \frac{L\alpha m}{2} + \frac{2\mu m}{1-\mu}$$

*This completes the proof.*

# B  DISTINCTION FROM NORMALIZED GRADIENT DESCENT

The proposed optimizer is fundamentally distinct from the Normalized Gradient Descent (NGD) algorithm (Nesterov, 1984; Cortés, 2006). While the update rule for scalar and vector parameters is identical to that of NGD, a significant divergence emerges when considering higher-dimensional parameters, particularly matrices.

For a matrix parameter $W$, the NGD update rule scales the gradient matrix $\nabla f(W)$ by its Frobenius norm, which is equivalent to a global learning rate adjustment. This approach preserves the directional information of the gradient unaltered. In contrast, our proposed optimizer not only modifies the learning rate but also applies a structural transformation to the gradient $\nabla f(W)$ itself. This transformation, which includes both a RACS transformation and a magnitude scaling, results in optimization dynamics that are fundamentally different from those of NGD.

Furthermore, we conducted an empirical evaluation to compare the performance of NGD under different learning rates on mathematical tasks. The learning rate was initialized with a 5% warm-up and subsequently decayed using a cosine schedule. The results, presented in Table 6, indicate a strong dependence of NGD on a large learning rate for optimal performance. Analysis of the train-

| Datasets | lr=3e-2 | lr=3e-3 | lr=3e-4 | lr=3e-5 |
|---|---|---|---|---|
| GSM8K (%) | 48.4 | **70.1** | 31.1 | 25.8 |
| MATH500 (%) | 57.4 | **60.2** | 53.0 | 24.8 |
| AIME24 (%) | 6.7 | 6.7 | **13.3** | 10.0 |
| Average (%) | 30.8 | **45.7** | 32.4 | 20.2 |

Table 6: Performance comparison of NGD with different learning rates across mathematical datasets.

ing loss revealed that at the large learning rates required for good performance, the loss exhibited minimal or no decrease during the initial training phase. This suggests that NGD is poorly suited for the fine-tuning of pre-trained models. One potential explanation is the high variance in the average magnitude of the NGD updates, which can lead to training instability when compared to optimizers like AdamW and our proposed method.

Specifically, for the update values $\tilde{M}$ in NGD, the RMS of the update is given by:

$$\text{RMS}(\tilde{M})^2 = \frac{1}{mn} \sum_{i=1}^{m} \sum_{j=1}^{n} \left( \frac{M_{ij}}{\|M\|_F} \right)^2 = 1.$$

This implies that the average magnitude for each entry of the matrix is fixed at $\frac{1}{\sqrt{mn}}$. For LLMs, the parameters of different layers (e.g., embedding layers, attention layers, and MLP layers) possess vastly different scales. This fixed magnitude for updates across varying layer scales makes it challenging for a single learning rate to appropriately adjust all matrix parameters.

In contrast, our proposed optimizer rescales the magnitude of each matrix according to its RMS norm and further balances the magnitudes of different channels within a matrix. These operations collectively contribute to a more stable training process. This distinction is particularly significant in the context of Large Language Models, where the vast majority of trainable parameters are represented by matrices. Consequently, the performance of our optimizer is expected to diverge significantly from that of NGD during the training of such models.

## C PRETRAINING EXPERIMENTS

In this section, we present the pretraining experiments conducted to evaluate the performance of the proposed regularized optimizer. We trained a Qwen-like model using various optimizers for comparative analysis. The training framework was adapted from the Moonlight repository, accessible at `https://github.com/MoonshotAI/Moonlight`.

The model architecture is a Qwen2-like configuration with a hidden size of 896, an intermediate size of 4864, 16 attention heads, and 12 hidden layers. The model was trained on the "Elriggs/openwebtext-100k" dataset for one epoch, consistent with the settings specified in the Moonlight framework. The training was performed with a learning rate of 1e-4 and 100 warm-up steps. The progression of the training loss is illustrated in Figure 2.

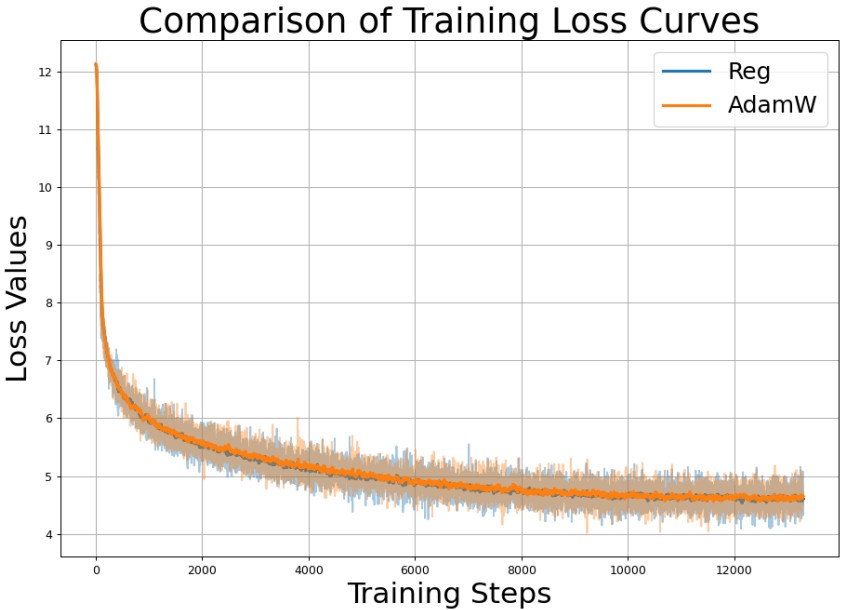

Figure 2: Training loss curves for AdamW and the regularized optimizer (REG) on the openwebtext-100k dataset.

As depicted in Figure 2, the training losses of the AdamW and the regularized optimizer are notably similar. This observation suggests that the regularized optimizer achieves a competitive performance level with AdamW, despite utilizing only the first momentum term.

# D  THE **REG** REGULARIZATION OPERATOR

In this study, the regularization operator, denoted as $\mathrm{reg}(\cdot)$, is defined as a normalization procedure applied to the update matrix $M$. Specifically, we employ a single pass of either row-wise or column-wise normalization.

The choice of this simplified operator warrants justification, particularly when contrasted with established practices in numerical analysis. It is common in the numerical analysis literature to iteratively apply row and column normalizations to a matrix to ensure desirable properties, like Ruiz normalization (Ruiz, 2001) or Pock-Chambolle normalization (Pock & Chambolle, 2011). However, our empirical results from SFT experiments indicate that a single normalization step is sufficient for effective regularization. We observed that applying additional normalization iterations yielded no significant performance gains and could adversely affect the training dynamics. Consequently, we adopt the most straightforward implementation that proved effective.

For the sake of completeness, we define a generalized version of the **REG** optimizer in Algorithm 1, which allows for multiple normalization iterations.

---

**Algorithm 1** The Generalized **REG** Optimizer

---

**Input:** Iteration count for normalization $t \in \mathbb{N}$, learning rate $\alpha > 0$, momentum hyperparameter $\mu \in (0, 1)$, norm order $p \in [1, +\infty]$, RMS-norm target $\rho_{\text{target}}$, weight decay $\lambda \geq 0$.

1: **for** each training step $k = 0, 1, 2, \ldots$ **do**
2:     // Compute update matrix with momentum
3:     $M_{k+1} \leftarrow \mu M_k + (1 - \mu)\nabla f(W_k)$
4:     // Apply iterative normalization
5:     $\tilde{M} \leftarrow M_{k+1}$
6:     **for** $i = 1, \ldots, t$ **do**
7:         // Row normalization
8:         $\tilde{M} \leftarrow \mathrm{diag}(\|\tilde{M}_{1,:}\|_p^{-1}, \ldots, \|\tilde{M}_{m,:}\|_p^{-1})\tilde{M}$
9:         // Column normalization
10:        $\tilde{M} \leftarrow \tilde{M} \, \mathrm{diag}(\|\tilde{M}_{:,1}\|_p^{-1}, \ldots, \|\tilde{M}_{:,n}\|_p^{-1})$
11:    **end for**
12:    // Scale to target RMS value
13:    $\hat{M}_{k+1} \leftarrow \tilde{M} \cdot \frac{\rho_{\text{target}}}{\mathrm{RMS}(\tilde{M})}$
14:    // Update weights with regularized gradient
15:    $W_{k+1} \leftarrow W_k - \alpha(\hat{M}_{k+1} + \lambda W_k)$
16: **end for**

---

The specific optimizer used throughout our experiments is a special case of Algorithm 1 where the normalization iteration count is set to $t = 1$. While values of $t > 1$ were found to yield marginal performance improvements, these gains were not statistically significant in our experimental setting.

