# OpenReview forum: "REG: A Regularization Optimizer for Robust Training Dynamics"
_ICLR.cc/2026/Conference — ICLR 2026 Conference Withdrawn Submission_

### Official Review · Reviewer_W97q · 2025-10-15

**Soundness:** 2
**Presentation:** 2
**Contribution:** 2
**Rating:** 4
**Confidence:** 4

**Summary:**

The paper proposes an optimizer called REG, which replaces the Muon's matrix-sign operator on the momentum with (Row-and-Column-Scaling) RACS operator, hypothesizing that the operation improves the conditioning of the matrix. Theoretically, the paper show that the subsequence generated by the optimizer is convergent. The optimizer is validated in the task of LLM finetuning and image classification, demonstrating improved downstream performance.

**Strengths:**

* The RACS operator is computationally cheap and is easy to implement.
* Under the distributed training setting, the communication cost of the algorithm is cheaper than Muon.
* REG exhibits improvements on the task of LLM finetuning and image classification, although the performance gap over the baseline methods is a bit marginal.

**Weaknesses:**

* **Motivation is not well-justified.** The motivation of REG's design is to improve the conditioning of the update (line 161-168). However, there seems no justification on why the $\text{normal}(\cdot)$ operator can achieve this. In fact, let $M=a b^\top$ be a rank-1 matrix, it is trivial to show that $\text{normal}(M; \rho)$ is also a rank-1 matrix, and the conditioning is not improved.

* **The evaluation of optimization capability is not comprehensive.** The paper does not provide the loss curve of LLM finetuning task, which is crucial in justifying the optimization capability of REG is relatively larger-scale experiments than image classification. The design of pre-training experiment in Appendix C is not very comprehensive. For instance, based on the work [1], AdamW is learning rate is set to approximately 1e-3, which is 10 $\times$ larger than the choice in this paper. It would be more convincing to compare REG with baseline methods under different learning rate. The comparison with Muon is missing.

[1] Muon is scalable for llm training.

**Questions:**

* If i understand it correctly, the convergence result in line 281 is for update (5), which essentially reduces to update (4) if we turn off the weight decay. Then, why update (4) asymptotically converges to local minimizer (Theorem 3) while the update (5) can only converge to the neighborhood?

---

### Official Review · Reviewer_CdYK · 2025-10-25

**Soundness:** 2
**Presentation:** 2
**Contribution:** 2
**Rating:** 2
**Confidence:** 4

**Summary:**

This paper proposes REG, a novel optimizer for stabilizing training dynamics of large language models (LLMs). REG replaces the matrix sign function in Muon with a Row-and-Column Scaling (RACS) operator that balances the gradient matrix by normalizing rows and columns. The method is theoretically motivated by matrix equilibration from numerical linear algebra and aims to provide smoother and more compatible update dynamics than Muon. Experiments on math reasoning, optimization modeling, and image classification show that REG performs comparably or slightly better than AdamW and much more stably than Muon.

**Strengths:**

1. Replacing the aggressive matrix sign with a softer normalization operator is a rational idea.
2. The paper tests REG across NLP and vision domains, showing its  generalizability
3. Analysis on theory is provided.

**Weaknesses:**

1.**Lack of large-scale experiments, results limited to small models, small datasets and SFT tasks.**

All experiments are conducted on small models limited-scale supervised fine-tuning. There is no evidence on pretraining large models (≥7B) on full corpora such as C4 (like galore that uses llama-7b architecture for pretraining from scratch ), where optimizer stability and efficiency truly matter. This severely limits the credibility of the claimed generality for LLM training. A convincing study would require at least one large-scale pretraining experiment demonstrating that REG maintains or improves convergence behavior at scale.

2. Accuracy average gains (≤1%) are small and statistically unclear (table 1), with no analysis of variance or ablation on convergence speed beyond simple tables.

3. The paper is algebra-heavy but lacks interpretive insight into optimizer behavior, and key messages are obscured by dense mathematics. More intuitive figures should be provided for potential readers.

**Questions:**

refer to Weaknesses

---

### Official Review · Reviewer_EGPF · 2025-10-28

**Soundness:** 3
**Presentation:** 3
**Contribution:** 2
**Rating:** 2
**Confidence:** 4

**Summary:**

This paper proposed new optimizer based on Row and Column Normalization (RACS) for fine-tuning tasks. The author propose to left or right multiply the momentum matrix with a diagonal re-scaling matrix depending on the dimensionality of the matrix. The motivation is that it should has less drastic regularization effect as Muon, which relies on matrix sign operation. Additionally, the author also uses a post-processing trick to make the RMS norm of the update constant, which is a well-known trick for optimization. Theoretically, the author proved the convergence of this REG when momentum is disabled, and provide an upper bound with momentum. But they do not provide theoretical advantages over existing optimizers.

**Strengths:**

The structure of this paper is solid and the content is easy to follow. The motivation is clear and the proposed algorithm is simple. It is also good to see some ablation studies to investigate some properties of REG. Although the theory is quite limited, it is still helpful to have a such convergence guarantee.

**Weaknesses:**

Despite that the proposed algorithm is simple and easy to understand, my main concern is that the contribution of this work is incremental. In fact, there are other previous work proposing similar procedures as REG, but these are not properly cited or compared to. For example, the well-known Adafactor, which also uses row and column l2 norm on $E[g^2]$ as scaling matrix. So the core difference compared to REG is that (1) Adafactor uses two-sided scaling; (2) REG uses momentum, whereas Adafactor uses EMA of $g^2$. However, REG is not compared with Adafactor in the empirical study. Even if REG shows better performance, the current content fails to explain why these differences gives better performance.

Apart from Adafactor, SinkGD [1] utilizes the Sinkhorn algorithm with l2 norm, which is the iterative version of REG with two-sides scaling, so REG with l2 norm is a special case of SinkGD, but the author fails to cite this and compare to it. Also, there is another RACS optimizer proposed in [2], which also proposing two-sided scaling solved by a fixed-point procedure. In fact, if one forces the $S$ and $Q$ matrix to be identity at the beginning, this is closely related to REG with l2 norm. But these work are not cited and compared to. Similarly, even REG gives better performance, the current content lacks the detailed explanation where these advantages comes from.


[1] Meyer Scetbon, et al. Gradient multi-normalization for stateless and scalable LLM training. In The Thirty-ninth Annual Conference on Neural Information Processing Systems (NeurIPS 2025).

[2] Gong, W., Scetbon, M., Ma, C., & Meeds, E. (2025). Towards Efficient Optimizer Design for LLM via Structured Fisher Approximation with a Low-Rank Extension. arXiv preprint arXiv:2502.07752.

**Questions:**

1. Major concerns are the in the weakness section.

2. In Line 43, you mentioned performance degradation. Do you have evidence for that? Because from [3] table 6, they uses Muon to finetune the model checkpoint trained by AdamW. It seems that there is no significant performance degradation. On some tasks, it performs even better than AdamW+AdamW. Can you explain this discrepancy?

3. What is the intuition behind which side the scaling matrix should be applied to? Why it depends on the dimension of the matrix?

4. Regularising optimizer's output to constant RMS norm is not new, both [3] and [4] uses similar trick.

5. Why choose one-side scaling in practice instead of two-sided scaling?

6. The performance of Muon in SFT experiment is surprisingly poor, which is not aligned with the findings from [3]. Can you explain why?


[3] Liu, J., Su, J., Yao, X., Jiang, Z., Lai, G., Du, Y., ... & Yang, Z. (2025). Muon is scalable for LLM training. arXiv preprint arXiv:2502.16982.
[4] Ma, C., Gong, W., Scetbon, M., & Meeds, E. (2024). SWAN: SGD with Normalization and Whitening Enables Stateless LLM Training. arXiv preprint arXiv:2412.13148.

---

### Official Review · Reviewer_rWMG · 2025-10-29

**Soundness:** 1
**Presentation:** 2
**Contribution:** 2
**Rating:** 2
**Confidence:** 5

**Summary:**

This work proposes to apply Row-and-Column-Scaling on AdamW to get a stabler optimizer for finetuning.

The idea could be reasonable. There is a related existing work [1] in ICML 2025 using max-norm, considering both column and row extreme values. The publication time of [1] could be quite close to the submission of this work, so a comparison is not necessary. The use of column norm and row norm is still novel to me to some extent.

[1] MERIT: Maximum-normalized Element-wise Ratio for Language Model Large-batch Training

**Strengths:**

The idea could be reasonable.

**Weaknesses:**

Currently, this work lacks a complete discussion of convergence. The experimental part is not solid enough, with inaccurate baselines. For example, in Table 1, Qwen2.5-Math-1.5B could have a baseline of around 60 on GSM8k; in Table 3, the SGD performance of ResNet on CIFAR-100 is far below normal.

Minor problems: In table 2, 84.66 and 37.00 are only bolded for REG while they have a tie.

**Questions:**

I would suggest that the authors improve their experiments with a solid baseline for comparison.

---

### Note · Authors · 2025-11-12

I have read and agree with the venue's withdrawal policy on behalf of myself and my co-authors.